# Calcein-Modified CeO_2_ for Intracellular ROS Detection: Mechanisms of Action and Cytotoxicity Analysis In Vitro

**DOI:** 10.3390/cells12192416

**Published:** 2023-10-07

**Authors:** Nikita N. Chukavin, Vladimir K. Ivanov, Anton L. Popov

**Affiliations:** 1Institute of Theoretical and Experimental Biophysics, Russian Academy of Sciences, Moscow 142290, Russia; chukavinnik@gmail.com; 2Scientific and Educational Center, State University of Education, Moscow 105005, Russia; 3Kurnakov Institute of General and Inorganic Chemistry of the Russian Academy of Sciences, Moscow 119991, Russia; van@igic.ras.ru

**Keywords:** cerium oxide nanoparticles, calcein, reactive oxygen species, antioxidant

## Abstract

Cerium oxide nanoparticles (CeO_2_ NPs) are metal-oxide-based nanozymes with unique reactive oxygen species (ROS) scavenging abilities. Here, we studied new CeO_2_ NPs modified with calcein (CeO_2_-calcein) as an intracellular ROS inactivation/visualization theranostic agent. The molecular mechanisms of the CeO_2_-calcein intracellular activity, allowing for the direct monitoring of ROS inactivation in living cells, were studied. CeO_2_-calcein was taken up by both normal (human mesenchymal stem cells, hMSc) and cancer (human osteosarcoma, MNNG/Hos cell line) cells, and was easily decomposed via endogenous or exogenous ROS, releasing brightly fluorescent calcein, which could be quantitatively detected using fluorescence microscopy. It was shown that the CeO_2_-calcein has selective cytotoxicity, inducing the death of human osteosarcoma cells and modulating the expression of key genes responsible for cell redox status as well as proliferative and migration activity. Such cerium-based theranostic agents can be used in various biomedical applications.

## 1. Introduction

Cerium oxide NPs are inorganic nanozymes capable of mimicking the activity of endogenous enzymes, such as catalase, superoxide dismutase, or peroxidase [1]. The catalytic properties of CeO_2_ NPs are based on the presence of cerium on their surface in two valence states: Ce^3+^/Ce^4+^, which makes redox reactions possible. The low energy of oxygen vacancy formation is the reason why CeO_2_ NPs easily participate in redox processes involving reactive oxygen species. 

CeO_2_ NPs have a high degree of biocompatibility, which has been confirmed via in vitro and in vivo studies. Dan et al. investigated the distribution and clearance of nanoparticles of different sizes and cerium ions introduced via intravenous infusion in rats. It has been shown that traditional pharmacokinetic models are best suited for cerium ions and 5 nm nanoparticles. CeO_2_ NPs with a size of 5 nm are quickly removed from the blood and do not cause toxic effects [2]. Yokel et al. studied the biodistribution of CeO_2_ NPs when administered intravenously to rats and showed that the nanoparticles that accumulated in the liver and spleen did not cross the blood–brain barrier, and caused no toxic effects [3].

CeO_2_ NPs show excellent potential for solving important biomedical problems, from regenerative medicine to cancer therapy. For example, in an earlier study, we showed that CeO_2_ NPs exhibit high biocompatibility with hMSc and primary mouse embryonic fibroblasts, and also significantly increase their proliferation activity [4,5,6]. It has also been shown that CeO_2_-NP-modified bacterial cellulose scaffolds increase the proliferative activity of mouse MSCs [7]. Citrate-stabilized CeO_2_ NPs exhibit good radioprotective activity [8]. This effect is not only due to the antioxidant activity of CeO_2_ NPs but also their ability to modulate the expression levels of genes associated with oxidative stress [9]. The antioxidant properties of CeO_2_ NPs provide efficient protection of primary mouse embryonic fibroblasts from plasma-induced oxidative damage [10]. CeO_2_ NPs restore the regenerative activity of planarians under the influence of low-intensity green light [11] or high doses of X-rays [12]. 

Reactive oxygen species play an important role in the processes of carcinogenesis and cancer therapy [13]. CeO_2_ NPs can be widely used in tumor redox therapy given their unique catalytic (enzyme-like) properties. The surface modification of CeO_2_ NPs makes it possible to impart new properties and modalities, such as pH sensitivity [14], GSH sensitivity [15], and hydrogen peroxide sensitivity [16]. For example, Li et al. designed new PEG-coated CeO_2_ NPs with chlorin-e6/folic-acid-loaded branched polyethylenimine (PPCNPs-Ce6/FA) for targeted photodynamic therapy to overcome drug-resistant breast cancers [17]. PPCNP-Ce6/FA generated reactive oxygen species after near-infrared irradiation, leading to reduced P-glycoprotein (P-gp) expression, lysosomal membrane permeabilization, and excellent phototoxicity toward resistant MCF-7/ADR cells. Thus, CeO_2_ NPs are considered a unique nanoplatform with a high degree of biocompatibility and unique catalytic properties, being very promising for a wide range of biomedical applications. 

Currently, there is significant interest in the design of new theranostic agents capable of combining diagnostics and therapy modalities, and cerium-containing nanomaterials show great promise for such applications. For example, CeO_2_ NPs provide strong X-ray attenuation due to cerium’s K-edge at 40.4 keV, and they can be used as a potential CT contrast agent [18,19]. The biocatalytic properties of CeO_2_ NPs can be further improved via doping with trivalent rare-earth elements, such as Gd, which has optimal properties for MRI imaging [20,21]. Ce-doped carbon dots exhibit excellent photoluminescent properties and can be applied in cell imaging. The antioxidant property of Ce-doped carbon dots alleviates the inflammatory stage in healing full-thickness excision wounds [22]. CeO_2_ NPs are a promising platform for nuclear medicine and tumor radiotherapy. Khabirova et al. proposed CeO_2_ NPs as nanoplatforms for radiopharmaceuticals containing radionuclides [23]. Radioactive CeO_2_ NPs were used to analyze cerium biodistribution and accumulation in the body of rats with various pathways of administration [24]. Thus, CeO_2_ NPs are a promising platform for cancer diagnosis and treatment due to their high biocompatibility, unique redox activity, and the possibility of their additional functionalization. 

We have previously demonstrated that the modification of CeO_2_ NPs with calcein results in an intracellular agent capable of ROS detection and inactivation in vitro [25]. However, the molecular mechanisms of the bioactivity of CeO_2_-calcein have not been previously disclosed, which we address in this paper. A comprehensive analysis of the molecular mechanisms of calcein-modified CeO_2_ NP bioactivity in normal and cancer cells in vitro was performed. The action of the nanoparticles on both normal and cancer cells was analyzed in view of their possible different efficiency in endocytosis of the nanoparticles, as well as their different intracellular metabolism and signaling pathways, which may also contribute to the activity and direction of the action of the nanoparticles.

## 2. Materials and Methods

### 2.1. CeO_2_-Calcein Synthesis and Analysis

The synthesis of calcein-modified CeO_2_ NPs was performed according to a recently reported procedure [25]. The microstructure of the calcein-modified CeO_2_ NPs was analyzed using transmission electron microscopy (TEM) on a Leo912 AB Omega electron microscope (Zeiss, Jena, Germany) at an accelerating voltage of 100 kV. Zeta potential measurements were performed using a Zetasizer Nano ZS analyzer (Malvern, United Kingdom) via diluting the calcein-modified CeO_2_ NPs in deionized water (mQ) at a temperature of 25 °C. The hydrodynamic diameter was measured in deionized water, Hanks’ buffer solution (PanEko, Moscow, Russia), in the DMEM/F12 culture medium (1:1) (PanEko, Moscow, Russia) and in the DMEM/F12 culture medium (1:1) containing 10% FBS (Biosera, Cholet, France) using a Beckman Coulter Particle Size Analyzer (N5 submicron) (Beckman, CA, USA).

### 2.2. Cell Culture

Human mesenchymal stem cells and human osteosarcoma cell culture MNNG/Hos were used. hMScs were isolated from the pulp of a third molar extracted for orthodontic reasons from a healthy 12-year-old patient (with his written consent). All the experiments were carried out in agreement with good clinical practice and the ethical principles of the current edition of the Declaration of Helsinki. The MNNG/Hos cell line is a certified human osteosarcoma cell line deposited in the Theranostics and Nuclear Medicine Laboratory of ITEB RAS. Cells were cultured in DMEM/F12 medium (1:1) (PanEco, Moscow, Russia) supplemented with L-glutamine (146 mg per 450 mL of medium) (PanEko, Moscow, Russia), penicillin, streptomycin (100× lyophilisate per 450 mL of medium) (PanEko, Moscow, Russia), and 10% fetal bovine serum (Biosera, Cholet, France). Cultivation was carried out in culture flasks with filter caps with a cultivation area of 25 and 75 cm^2^ (TPP, Trasadingen, Switzerland) in a CO_2_ incubator at a temperature of 37 °C.

### 2.3. MTT Assay

The viability of the cells was analyzed using an MTT assay. This method is based on the phenomenon of the reduction of water-soluble MTT salt (3-(4,5-dimethylthiazol-2-yl)-2,5-diphenyl-tetrazolium bromide) (PanEko, Moscow, Russia) to purple water-insoluble formazan via cellular NAD(P)H-dependent oxidoreductases. After 24, 48, and 72 h of cell incubation in the presence of the calcein-modified CeO_2_ NPs, the culture medium was replaced with an MTT solution (0.5 mg/mL) in the medium without serum. After 3 h of incubation, the medium with MTT residues was removed, and DMSO (PanEko, Moscow, Russia) was added to the wells of the plate. The plates were then placed on a plate shaker to mix the contents of the wells and dissolve the resulting formazan in DMSO. After 10 min of mixing, the absorbance of the solutions was measured at a 540 nm wavelength using a Bio-Rad 680 plate spectrophotometer (Bio-Rad, Hercules, CA, USA). The absorbance values were converted to percentages of the values of the control groups, and the deviations in the samples were indicated as standard deviation (SD).

### 2.4. Live/Dead Assay

The cytotoxic effect of the calcein-modified CeO_2_ NPs was assessed using a Live/Dead assay. This method was used to calculate the percentage ratio of the number of cells that died during incubation with the calcein-modified CeO_2_ NPs to their total number. Cell numbers were counted using fluorescence microscopy, and cells were labeled via staining with a combination of the fluorescent dyes Hoechst 33,342 (binds to DNA of all cells, excitation wavelength 350 nm, emission wavelength 460 nm) and propidium iodide (binds to DNA of dead cells, excitation wavelength 535 nm, emission wavelength 615 nm). After 24, 48, and 72 h of incubation, the culture medium was replaced with a mixture of Hoechst 33,342 fluorescent dyes and propidium iodide in Hanks’ buffer solution (PanEko, Moscow, Russia). After 15 min of incubation with dyes, the cells were washed three times with Hanks’ buffer solution and were further analyzed using a ZOE imager (Bio-Rad, USA). ImageJ software was used to count the number of cells. Three areas on the field were analyzed on three different microphotographs. Quantitative analysis was presented as mean ± SD.

### 2.5. Wound-Healing Assay

Cell migration activity was analyzed using the wound-healing assay. Cell suspensions were placed in silicone inserts with 2 wells (Ibidi, Berlin, Germany) attached to the surface of the wells of a 24-well plate. Then, 100 µL of hMSc and MNNG/Hos cell suspensions was added to each well in the amount of 4 × 10^4^ and 2 × 10^4^ cells, respectively. After cell attachment, the medium in the inserts was replaced with the culture medium containing calcein-modified CeO_2_ NPs at concentrations of 10 μM, 100 μM, and 1 mM for subsequent overnight incubation. After incubation, the remains of the medium with calcein-modified CeO_2_ NPs were removed from the well, together with the insert, then 400 μL of the culture medium was added to the well, and the cells were left for cultivation for 48 h. During cultivation, the gap area between cells was photographed using a CloneSelect Imager system (Molecular Devices, San Jose, CA, USA), and images were taken every 12 h. At the end of cultivation, the dynamics of the reduction in the gap area between cells were analyzed using ImageJ software.

### 2.6. Real-Time Polymerase Chain Reaction (RT-PCR)

The effect of the calcein-modified CeO_2_ NPs on the expression of genes associated with oxidative stress was analyzed using RT-PCR. The hMSc and MNNG/Hos cell lines were seeded into a 6-well plate at a seeding density of 7 × 10^4^ and 3 × 10^4^ cells/cm^2^, respectively. After cell attachment, the culture medium was replaced with calcein-modified CeO_2_ NPs at a concentration of 10 µM or 1 mM. After 24 h of incubation, the cells were removed from the plate and transferred to a suspension in an EverFresh solution (Silex, Moscow, Russia) with a volume of 500 µL for storage at −20 °C. Then, full-length poly-(A) mRNA was isolated from the obtained samples of cell suspensions on magnetic particles (Sileks, Moscow, Russia). Next, the first strand of cDNA was synthesized from the isolated mRNA using an MMLV RT kit (Evrogen, Moscow, Russia). The resulting cDNA was amplified during qPCR using a qPCRmix-HS SYBR + LowROX kit (Evrogen, Moscow, Russia) and a primer set for human genes associated with oxidative stress (Appendix A). GAPDH, actin, and RPLP0 genes were used as housekeeping reference genes. The processes of mRNA isolation, cDNA synthesis, and qPCR were carried out in strict accordance with the protocols specified by the manufacturers. cDNA synthesis was performed using a Bio-Rad C1000 cycler, and qPCR was performed using a QuantStudio™ 5 Real-Time PCR System (ThermoFisher, Waltham, MA, USA). The qPCR results were processed using Genesis software, and differences in gene expression levels between the samples were analyzed using principal component analysis. Three different samples were used, with a single repeat per sample.

### 2.7. Antioxidant Activity Analysis of Calcein-Modified CeO_2_ NPs

The antioxidant activity of the calcein-modified CeO_2_ NPs was studied under oxidative stress conditions induced via hydrogen peroxide (1 mM for 1 h) by assessing the membrane mitochondrial potential (MMP) and the level of reduced glutathione (GSH) via staining cells with tetramethylrhodamine (TMRE) fluorescent dyes (Invitrogen, Carlsbad, CA, USA) (excitation wavelength 550 nm, emission wavelength 575 nm) and ThiolTracker Violet (Invitrogen, Carlsbad, CA, USA) (excitation wavelength 405 nm, emission wavelength 525 nm). To perform the MTT assay, Live/Dead assay, and analysis of the bioactivity of calcein-modified CeO_2_ NPs in the oxidative stress model, hMSc and MNNG/Hos cells were seeded in 96-well plates (SPL, Pyeongtaek, South Korea) at a density of 4 × 10^4^ and 2.5 × 10^4^ cells/cm^2^, respectively. Subsequently, the cells were incubated with calcein-modified CeO_2_ NPs at various concentrations in a culture medium supplemented with 10% FBS. After 24 h of cell incubation, the cells were stained with fluorescent dyes, and 1 mM hydrogen peroxide solution was added. Then, the dynamics of dye fluorescence intensity in cells was analyzed using a Biotek Synergy H1 (Agilent, Santa Clara, CA, USA) plate reader over the 3 h of incubation with H_2_O_2_ solution. Measurements were carried out while maintaining a temperature of 37 °C and 5% CO_2_ concentration. After 24 h of cell incubation in the presence of the nanoparticles, the cells were washed, and a 1 mM hydrogen peroxide solution in Hanks’ buffer solution was added. Then, the dynamics of calcein fluorescence intensity in cells was analyzed using a Biotek Synergy H1 plate reader over the 3 h of incubation with a H_2_O_2_ solution. Measurements were carried out while maintaining a temperature of 37 °C and 5% CO_2_ concentration in the tablet reader. Additionally, the release of calcein from the nanoparticles in cells was visualized in an oxidative stress model using a ZOE cell imager (Bio-Rad, USA). Cells were photographed before, immediately after, and 1 h after the addition of 1 mM H_2_O_2_ solution.

### 2.8. Statistical Analysis

The data were analyzed using GraphPad Prism 8 software. The statistical significance of the deviations between the test sets and the control was confirmed using the Welch *t*-test at 0.01 < *p* < 0.05 (*), 0.001 < *p* < 0.01 (**), 0.0001 < *p* < 0.001 (***), and *p* < 0.0001 (****). Images were processed using ImageJ and Adobe Photoshop CC (2017) software.

## 3. Results and Discussion

### 3.1. The Analysis of Calcein-Modified CeO_2_ NPs

In the calcein-modified CeO_2_ NPs, calcein molecules are adsorbed on the nanoparticle surface (Figure 1a). The size of the CeO_2_ NPs according to TEM data is 3–4 nm (Figure 1b). The UV-visible spectrum of the calcein-modified CeO_2_ NPs confirms the presence of two characteristic peaks of cerium oxide and calcein (Figure 1c). The hydrodynamic diameter of the calcein-modified CeO_2_ NPs in MQ water is about 5–7 nm (Figure 1d), and the zeta potential is −26.7 mV. The stability of the calcein layer under the various ionic microenvironments was analyzed additionally (Appendix A). It was shown that sulfate (SO_4_^2−^), chloride (Cl^−^), and nitrate (NO_3_^−^) species do not trigger calcein desorption from the surface of the CeO_2_ NPs. Under selected conditions, only hydrogen peroxide was able to replace calcein from the surface complex and activate its fluorescence.

### 3.2. Cytotoxicity Analysis of the Calcein-Modified CeO_2_ NPs

We performed a comprehensive analysis of cytotoxicity and determined that the calcein-modified CeO_2_ NPs do not suppress the viability of hMSc and MNNH/Hos cells across a wide range of concentrations (from 10 µM to 1 mM; Figure 2 and Figure 3). At the same time, the highest concentration of nanoparticles (10 mM) leads to a pronounced inhibition of cellular metabolic activity. The most pronounced cytotoxic effects of the nanoparticles are manifested for the cancer cells of the MNNG/Hos line (Figure 2 and Figure 3). It should be noted that after 72 h of incubation of hMSc with the nanoparticles at a 10 mM concentration, the proportion of dead cancer cells was 36%. 

The selective anticancer effect of CeO_2_ NPs was reported earlier in a number of papers. In particular, Zhang et al. have shown that polyvinylpyrrolidone-stabilized CeO_2_ NPs selectively protect human normal cells but not cancer cells from ROS damage upon exposure to UV-radiation, suggesting their potential applications in photodynamic therapy (PDT) [26]. Redox-active CeO_2_ NPs selectively kill A375 melanoma cells through increasing intracellular ROS levels via mitochondrial dysfunction [27]. Polyvinylpyrrolidone-stabilized CeO_2_ NPs were shown to affect curcumin cytotoxicity and photocytotoxicity, depending on cell type, being more toxic to cancer cells in a selective manner [28]. A selective cytotoxic effect of CeO_2_ NPs on MCF-7 tumor cells was revealed when combined with paclitaxel, a widely used chemotherapeutic drug [29]. The combination of CeO_2_ NPs and paclitaxel led to a decrease in cell adhesion, migration, and viability. The mechanisms of selective action of CeO_2_ NPs may be associated with a change in catalytic activity under acidic microenvironmental conditions, as was previously shown in a model of chemotherapy-induced acute kidney injury [30]. Our data confirm the selective cytotoxicity of the calcein-modified CeO_2_ NPs.

### 3.3. Cell Migration Analysis

Cell migration activity is one of the key indicators of their viability [31]. For example, for human MSCs, this parameter is an indicator of regenerative potential, and for cancer cells, it is an indicator of potential metastatic activity [32]. Thus, we explored the effect of high concentrations of calcein-modified CeO_2_ NPs on the migration activity of cells after 24 h of incubation. It was found that a 0.1 mM concentration increased the migration activity of hMSc cells by 5.9% and MNNG/Hos cells by 23.1% (Figure 4). The reasons for the acceleration of the migration activity of osteosarcoma cells require additional research to clarify possible issues in the biosafety of ceria-based nanomaterials.

### 3.4. Antioxidant Activity of the Calcein-Modified CeO_2_ NPs

We further analyzed the antioxidant activity of the calcein-modified CeO_2_ NPs when cells were exposed to hydrogen peroxide in vitro. The mitochondrial membrane potential under oxidative stress was analyzed using the potential-sensitive fluorescent TMRE dye, which is able to penetrate into mitochondria and fluoresce, depending on the charge of the inner mitochondrial membrane (Figure 5a,b). It was found that calcein-modified CeO_2_ NPs at concentrations of 0.01–1 mM increase the MMP of hMSc and MNNG/Hos cell types both under standard and oxidative stress conditions. The nature of the increase in mitochondrial membrane potential (MMP) depends on the concentration of the nanoparticles, cell lines, and experimental conditions. Under standard conditions, the action of the nanoparticles on the cells leads to a concentration-dependent increase in their MMP: by 10–16% for hMSc cells and by 7–23% for MNNG/Hos cells. Meanwhile, under oxidative stress conditions, the calcein-modified CeO_2_ NPs exhibit their activity in an extremely specific way with respect to different cell lines. In the absence of the calcein-modified CeO_2_ NPs, the MMP of both cell types decreases by 17%. At the same time, the MMP of hMSc cells under the action of the nanoparticles does not significantly change upon H_2_O_2_ exposure; meanwhile, for the MNNG/Hos cell line, a concentration-dependent increase in MMP is observed up to the values exceeding the untreated control under standard conditions by 10.8%. The level of non-enzymatic antioxidants was also analyzed using the fluorescent dye ThiolTracker, which binds to reduced glutathione inside cells in direct proportion to its concentration, which makes it possible to assess the redox status of cells. It was shown that the preincubation of the cells with the nanoparticles reduces the level of reduced glutathione (GSH) in normal and cancer cells (Figure 5c,d). Under standard conditions, the GSH level decreases by 13% in hMSc cells and by 8.5% in MNNG/Hos cells when the concentration of the nanoparticles reaches 1 mM. When the cells were exposed to hydrogen peroxide, an increase in the level of GSH was observed in the hMSc cells (by 31%) and in the MNNG/Hos cells (by 41.2%). When the cells were pretreated with calcein-modified CeO_2_ NPs, the level of GSH in hMSc decreased by 18.4–32% under standard conditions. In cancer cells, the GSH level in the presence of the nanoparticles decreased by 15.6–32.7%, approaching the control level under standard conditions (Figure 5c,d).

### 3.5. Functional Activity of the Calcein-Modified CeO_2_ NPs

To analyze the functional activity of calcein-modified CeO_2_ NPs, we studied the dynamics of calcein fluorescence intensity after cell treatment with hydrogen peroxide (Figure 6a–c). The binding of calcein via the CeO_2_ NPs leads to the quenching of its fluorescence. Meanwhile, hydrogen peroxide, having an increased affinity for the surface of CeO_2_ NPs, causes the release of calcein from the surface of the nanoparticles when it interacts with the peroxide and restores the fluorescence of the dye. As shown in Figure 6d, the incubation of the cells in the presence of the nanoparticles leads to a concentration-dependent accumulation of free fluorescent calcein in cells. The induction of oxidative stress in cells enhances the process of calcein release from the surface of the nanoparticles, which is schematically shown in Figure 6e. At the same time, extremely weak calcein fluorescence is observed in hMSc cells compared to that in MNNG/Hos cells. It was previously shown that the oxidative stress induced via H_2_O_2_ leads to a decrease in MMP and an increase in the expression of genes for gamma-glutamylcysteine synthetase (gamma-GCS), glutathione synthetase, and glutathione reductase, which is accompanied by an increase in the level of GSH [33]. We have shown that the action of the calcein-modified CeO_2_ NPs simultaneously leads to an increase in MMP and a decrease in the level of GSH in cells. This effect manifests itself regardless of the experimental conditions, both without and with H_2_O_2_. We assume that the bioactivity of calcein-modified cerium oxide nanoparticles is either not associated with the redox activity of CeO_2_ NPs or is only partially associated with it, and is primarily due to its effect on various molecular cascades in the cells.

Thus, the synthesized calcein-modified CeO_2_ NPs have the ability to detect and simultaneously inactivate exogenous hydrogen peroxide in cells across a wide range of concentrations.

### 3.6. Gene Expression Analysis

Calcein-modified CeO_2_ NPs change the expression patterns of genes associated with oxidative stress. Bioactivity varies depending on the concentration of the nanoparticles and the cell type (Figure 7a,b). 

#### 3.6.1. Glutathione Peroxidase Gene Family Analysis

For the genes of peroxidases and S-transferases, the following phenomena are observed: the expression of the GPX2 and GPX3 genes increases in both normal and cancer cells with an increase in the concentration of the nanoparticles; GPX4 gene expression somewhat increases in both cell lines, but only at a nanoparticle concentration of 10 μM; the expression of the GPX5 gene increases in MNNG cells in a concentration-dependent manner, while in MSCs it decreases at a 10 μM nanoparticle concentration and increases at a 1 mM concentration; and GSTP1 glutathione-S-transferase gene expression increases only in hMSc cells and only at low nanoparticle concentrations, while GSTZ1 gene expression decreases in normal cells and increases in cancer cells at a nanoparticle concentration of 1 mM. 

#### 3.6.2. Peroxiredoxin Gene Family Analysis

The following phenomena are observed for the peroxiredoxin cluster genes: the expression of the PRDX1, PRDX2, PRDX3, PRDX4, PRDX5, and PRDX6 genes in the presence of the nanoparticles increases in a concentration-dependent manner in the MNNG/Hos cell line, while in hMSc cells, their expression somewhat decreases at a nanoparticle concentration of 10 μM and increases at a concentration of 1 mM. 

#### 3.6.3. Peroxidase Gene Family Analysis

For the genes of antioxidant enzymes, the following phenomena are observed: the expression of the catalase (CAT) gene is increased only in cancer cells at a nanoparticle concentration of 1 mM; the expression of the CYBB gene (the subunit of NOX2 NADH oxidase) in the presence of the nanoparticles increases in a concentration-dependent manner in MNNG/Hos cells, while in hMSc cells, its expression decreases at a 10 μM nanoparticle concentration and increases at a 1 mM concentration; the expression of the cytoglobin gene (CYGB) in the presence of the nanoparticles increases in a concentration-dependent manner in both cell lines; the expression of gene homologs of double oxidase (DUOX1 and DUOX2) in the presence of the nanoparticles increases in a concentration-dependent manner in cancer cells, while in normal cells, the expression of DUOX1 increases with an increase in the concentration of nanoparticles and the expression of DUOX2 decreases with a decrease in nanoparticle concentrations; and the expression of lactoperoxidase (LPO) and myeloperoxidase (MPO) genes increases with an increase in the concentration of the nanoparticles in both cell lines, and this pattern is most pronounced in cancer cells. In the presence of the nanoparticles, the expression of the PTGS1 and PTGS2 genes for cyclooxygenases increases in a concentration-dependent manner in MNNG/Hos cells, while in the hMSc cells, the expression of PTGS1 increases, and the expression of PTGS2 decreases in the presence of the nanoparticles, regardless of their concentration. 

#### 3.6.4. Antioxidant Gene Family Analysis

The cellular antioxidant cluster genes, which are not related to enzymes, are characterized by the following changes: the expression of the albumin (ALB) gene increases in the presence of the nanoparticles with an increase in their concentration in both cell lines, and this effect is more pronounced in cancer cells; the expression of the apolipoprotein E (APOE) gene decreases in cancer cells at a 1 mM nanoparticle concentration and increases in normal cells, regardless of calcein-modified CeO_2_ NP concentration; the expression of the glutathione reductase (GSR) gene increases in MNNG/Hos cells at a 1 mM nanoparticle concentration; the expression of the metallothionein (MT3) gene decreases in both cell lines at a 10 μM nanoparticle concentration, while it increases in cancer cells at a 1 mM nanoparticle concentration; the expression of the sulfiredoxin-1 (SRXN1) gene in the presence of the nanoparticles increases in a concentration-dependent manner in both cell lines; and the expression of superoxide dismutase genes (SOD1, SOD2, and SOD3) increases in MNNG/Hos cells in the presence of the nanoparticles, while in hMSc cells, SOD1 expression decreases with increasing nanoparticle concentrations, SOD2 expression increases and SOD3 expression decreases at a 10 µM nanoparticle concentration, and increases at a nanoparticle concentration of 1 mM.

#### 3.6.5. Analysis of Genes Involved in ROS Metabolism

For genes involved in ROS metabolism, the following changes in expression patterns are observed: the expression of a wide range of genes in cancer cells (12-lipoxygenase (ALOX12), inducible nitric oxide synthase 2 (NOS2), NOX2 paralogs, NOX4 and NOX5 proteins, heat shock protein 1 (HSPA1A), BCL2/adenovirus E1B 19 kDa protein-interacting protein 3 (BNIP3), epoxide hydrolase 2 (EPHX2), MPV17 protein, antioxidant protein 1 (ATOX1), 24-dehydrocholesterol reductase (DHCR24), transcription factor FOXM1, ferritin heavy chain protein (FTH1), glutamate-cysteine ligase regulatory subunit protein (GCLM), glutatione synthetase (GSS), heme oxygenase (HMOX), aldehyde oxidase 1 (AOX1), mannose-binding lectin (MBL2), NAD(P)H-dehydrogenase (quinone 1) (NQO1), protein RNF7, NAD-dependent deacytylase sirtuin 2 (SIRT2), and sequestosome 1 (SQSTM1)) increases with increasing nanoparticle concentrations; the expression of the mitochondrial uncoupler 2 (UCP2) gene increases in hMSc cells and decreases in MNNG/Hos cells at a 1 mM nanoparticle concentration; the expression of the chemokine ligand 5 (CCL5) gene in normal cells increases at a 10 μM nanoparticle concentration and decreases at a 1 mM nanoparticle concentration, while in cancer cells, the opposite pattern is observed; in normal cells, the expression of the ALOX12, NOX5, HSPA1A, EPHX2, FTH1, FOXM1, HMOX1, MBL2, SIRT2, and SQSTM1 genes increases in the presence of the nanoparticles, while the expression of the NOX4, BNIP3, MPV17, GCLM, and AOX1 genes decreases; the expression of the NOS2 and DHCR24 genes in the hMSc cell line increases with an increase in nanoparticle concentrations and decreases with a decrease in their concentration, while for the expression of the ATOX1, NQO1, and RNF7 genes, the opposite pattern is observed.

#### 3.6.6. Pathway Signature Gene Analysis

The following changes are observed for the genes encoding the signature participants of various signaling pathways: the expression of the genes for aldoketoreductase AKR1C2, the chaperone-regulator BAG2, the FHL2 protein, alpha-galactosidase A GLA, and the heat shock protein HSP90AA1 in cancer cells increases with increasing nanoparticle concentrations, while TRAPPC6A protein expression is reduced under these conditions; the expression of the BAG2 and FHL2 genes in the hMSc cell line decreases with increasing nanoparticle concentrations, while the expression of the LHPP and TRAPPC6A genes increases; in normal cells, AKR1C2 gene expression decreases at a 1 mM nanoparticle concentration and increases at a 10 μM concentration, while LHPP expression in cancer cells changes in the opposite direction. 

#### 3.6.7. Mitochondrial Dysfunction Gene Analysis

For genes whose expression disturbance mediates mitochondrial dysfunction, the following changes are observed: in the MNNG/Hos cell line, with an increase in nanoparticle concentrations, the expression of a number of genes increases, including genes for the ML43 protein of the large 39S subunit of the mitochondrial ribosome (MRPL43), subunit 11 of the mitochondrial NADH-dehydrogenase complex (NDUFB11), DNA-dependent mitochondrial RNA polymerase (POLRMT), sirtuin 1 (SIRT1), sirtuin 3 (SIRT3), mitochondrial transcription factor A (TFAM), paralogs of mitochondrial dimethyladenosine transferase (TFB1M and TFB2M), copper chaperone SOD1 (CCS), and selenoprotein S (SELENOS); in normal cells, the expression of the MRPL43, NDUFB11, and SELENOS genes somewhat increases in the presence of the nanoparticles at a 1 mM concentration and decreases at a concentration of 10 μM; the expression of the POLRMT gene in the hMSc cell line increases with an increase in the concentration of the nanoparticles; the expression of the TFB1M, TFB2M, and CCS genes decreases in normal cells with an increase in nanoparticle concentrations. The expression pattern of the SIRT1, SIRT3, and SIRT6 genes in hMSc cells is fundamentally different from that in MNNG/Hos cells: in normal cells, SIRT1 gene expression decreases with decreasing nanoparticle concentrations, SIRT3 gene expression increases at high nanoparticle concentrations and decreases at low concentrations, and SIRT6 gene expression increases in the presence of a 10 μM nanoparticle concentration, while in cancer cells, it decreases, regardless of their concentration; TFAM gene expression in normal cells increases with an increase in nanoparticle concentrations and decreases with a decrease in nanoparticle concentrations. 

#### 3.6.8. Anti-Apoptotic Gene Analysis

The genes whose products prevent the development of apoptosis in the cell are characterized by the following changes: the expression of the BCL2 protein gene in the presence of the nanoparticles increases in a concentration-dependent manner in both cell lines; the expression of the protein gene containing the baculovirus IAP repeat, BIRC3, decreases in normal cells with a decrease in nanoparticle concentrations, while in cancer cells, its expression slightly increases at a 1 mM nanoparticle concentration and decreases at a 10 μM concentration; and the expression of the factor 2 gene associated with the TNF receptor, TRAF2, increases in hMSc cells regardless of nanoparticle concentrations and decreases in MNNG/Hos cells in the presence of a 1 mM nanoparticle concentration. 

#### 3.6.9. Autophagy Gene Analysis

For genes whose products promote the development of cell autophagy, the following changes are observed: the expression of genes for autophagy-associated proteins (ATG3 and ATG12) and beta-1 kinase of ribosomal protein S6 (RPS6KB1) in cancer cells increases in the presence of a 1 mM nanoparticle concentration, while the gene expression of NfKb subunit (NFKB1) increases in the presence of the nanoparticles, regardless of their concentration; in normal cells, the expression of the ATG12, NFKB1, and RPS6KB1 genes decreases in the presence of the nanoparticles at various concentrations, while the expression of ATG3 increases with an increase in their concentration and decreases with its decrease. 

#### 3.6.10. Necrosis Gene Analysis

For the genes associated with cell necrosis, the following changes are observed: in both cell lines, the expression of the CCDC103, FOXI1, and RAB25 genes increases in the presence of the nanoparticles, and in cancer cells, their expression increases with an increase in the concentration of the nanoparticles, while in normal cells, the expression of CCDC103 increases regardless of their concentration, FOXI1 expression increases with increasing concentrations, and RAB25 expression slightly increases only at a 1 mM nanoparticle concentration; JPH3 gene expression increases in normal cells at a 10 μM nanoparticle concentration and decreases in cancer cells at a 1 mM concentration.

#### 3.6.11. Pro-Apoptotic Gene Analysis 

Proapoptotic genes are characterized by the following changes: in the MNNG/Hos cell line, the expression of genes for the Bcl-2-associated X (BAX) protein, the 40 CD40 differentiation cluster, the apoptosis regulator CFLAR, and the Fas receptor FAS increases in a concentration-dependent manner in the presence of the nanoparticles; a similar pattern is observed in hMSc cells for the CD40 gene; in normal cells, the expression of the BAX gene increases with a decrease in the concentration of the nanoparticles, while the exact opposite pattern is observed for the CFLAR gene; and in the hMSc cell line, with an increase in the nanoparticles concentration, the expression of the FAS gene decreases.

An analysis of the differences in the expression patterns of genes associated with oxidative stress in normal and cancer cells in the presence of the nanoparticles was carried out using the principal component method and showed the following: in cancer cells of the MNNG/Hos line, in the presence of the nanoparticles, there is a tendency for a concentration-dependent increase in the expression of the gene pattern, and moreover, the effect of a 1 mM nanoparticle concentration is much more pronounced than that of a 10 μM nanoparticle concentration. In normal cells of the hMSc line in the presence of the nanoparticles, a different picture is observed: in some cases, different concentrations of the nanoparticles alter gene expression differently, and the effect of a 10 μM nanoparticle concentration is slightly inferior to the effect a 1 mM nanoparticle concentration. It should be noted that the greatest effect from the action of the nanoparticles is observed in cancer cells at their highest concentration (Figure 7c). 

Thus, it can be concluded that calcein-modified CeO_2_ NPs not only significantly affect the expression pattern of genes associated with the redox status of the cell but also trigger signaling pathways in the processes of cell proliferation and differentiation.

## 4. Conclusions

CeO_2_ NPs are unique inorganic antioxidants with a high degree of biocompatibility and redox activity. The additional functionalization of CeO_2_ NPs provides them with new unique properties and possibilities, which opens up new directions for biomedical applications. In this research, we have studied the biocompatibility and molecular mechanisms of the bioactivity of calcein-modified CeO_2_ NPs, which are able to simultaneously visualize and inactivate ROS in the cell, on normal and cancer cells in vitro.

It was shown that the calcein-modified CeO_2_ NPs have an ultra-small size (4–5 nm), possess high colloidal stability, and have the ability to simultaneously inactivate and visualize the intracellular ROS location. Both normal and cancer cells actively uptake calcein-modified CeO_2_ NPs, with osteosarcoma cells exhibiting more effective uptake. Calcein-modified CeO_2_ NPs do not demonstrate cytotoxicity against normal and cancer cells across a wide range of concentrations (from 10 µM to 1 mM). However, high concentrations of calcein-modified CeO_2_ NPs (10 mM) lead to cell death, and the greatest cytotoxic effect is observed in cancer cells. We assume that the cytotoxicity of calcein-modified CeO_2_ NPs at high concentrations may arise due to the excessive accumulation of free calcein in cells. In particular, calcein can disrupt the ionic balance of the cell via binding Ca^2+^ and Mg^2+^ ions. The action of calcein-modified CeO_2_ NPs simultaneously leads to an increase in MMP and a decrease in the level of GSH in cells. This effect manifests itself regardless of the experimental conditions, both with and without H_2_O_2_ addition. We assume that the effect of the bioactivity of the nanoparticles is either not associated with the redox activity of CeO_2_ NPs or is only partially associated with it, and is primarily due to their effect on various molecular cascades in the cell.

We believe that the study of the bioactivity of calcein-modified CeO_2_ NPs will reveal the molecular mechanisms of the bioactivity of cerium-containing nanomaterials, thereby promoting the development of new non-toxic theranostic agents to achieve therapeutic and diagnostic effects.

## Figures and Tables

**Figure 1 cells-12-02416-f001:**
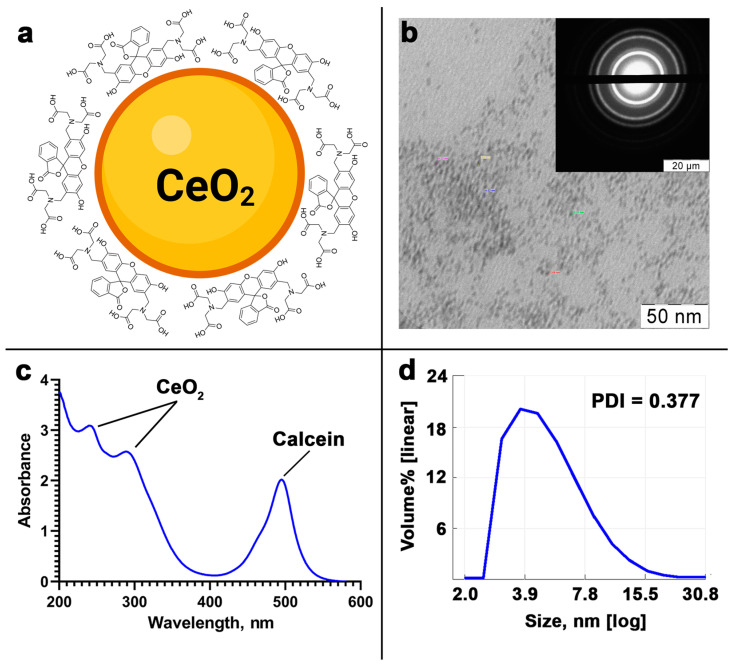
The schematic structure of calcein-modified CeO_2_ NPs (**a**), TEM image with the corresponding selected area diffraction pattern (SAED) (**b**), UV-visible spectrum (**c**), and the hydrodynamic diameter (**d**) of the calcein-modified CeO_2_ NPs.

**Figure 2 cells-12-02416-f002:**
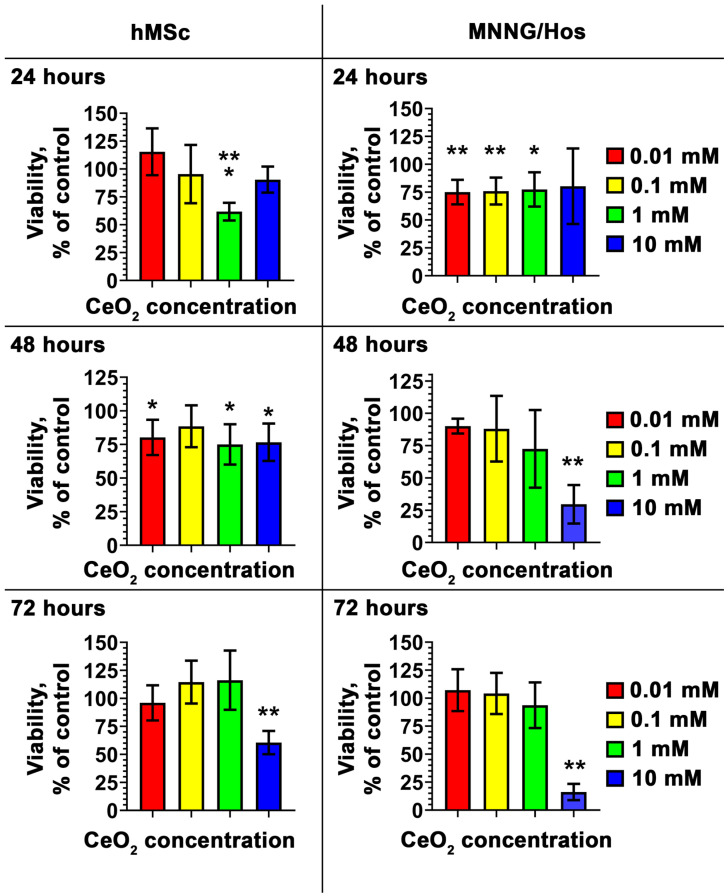
Viability of hMSc and MNNG/Hos cells 24, 48, and 72 h after incubation with calcein-modified CeO_2_ NPs at various concentrations (0.01–10 mM) via MTT assay. Data are presented as standard deviation (SD). The statistical significance of deviations between the control and test samples is indicated by *, where * *p* < 0.05, ** *p* < 0.01 and *** *p* < 0.001.

**Figure 3 cells-12-02416-f003:**
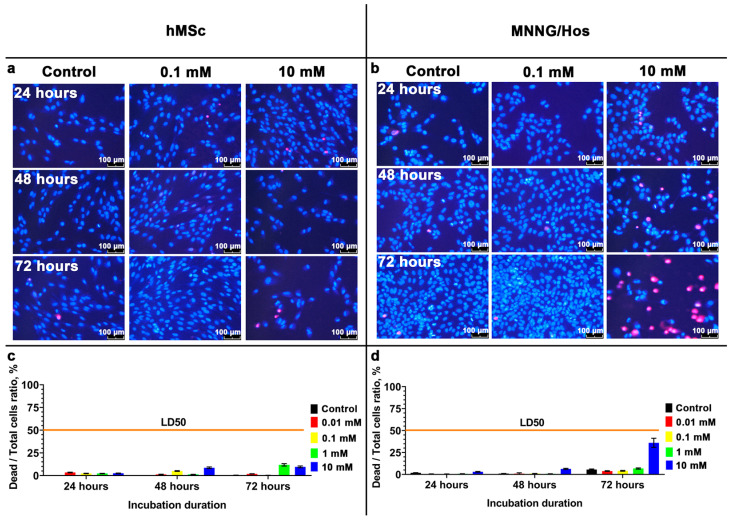
Live/Dead assay of hMSc and MNNG/Hos cells 24, 48, and 72 h after incubation with calcein-modified CeO_2_ NPs at various concentrations (0.01–10 mM). Cells were stained with fluorescent dyes Hoechst 33,342 (stains the nuclei of all cells blue) and propidium iodide (stains the nuclei of dead cells red) (**a**,**b**), and quantitative analysis of dead cells (**c**,**d**). The IC50 line corresponds to a value of 50% dead cells. The data on the c and d panels are presented as mean ± SD.

**Figure 4 cells-12-02416-f004:**
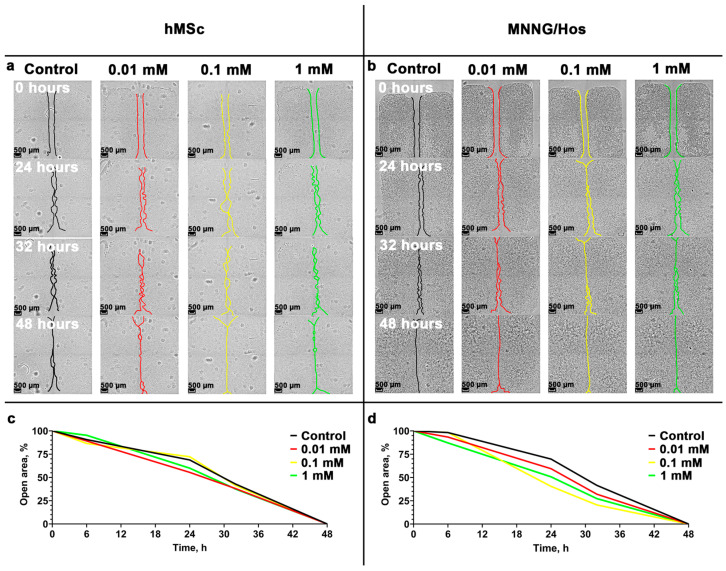
Wound-healing assay of hMSc and MNNG/Hos cells during incubation with calcein-modified CeO_2_ NPs at various concentrations (0.01–10 mM) (a,b), and cell migration dynamics (**c**,d). Cell microphotographs were obtained at 20x magnification.

**Figure 5 cells-12-02416-f005:**
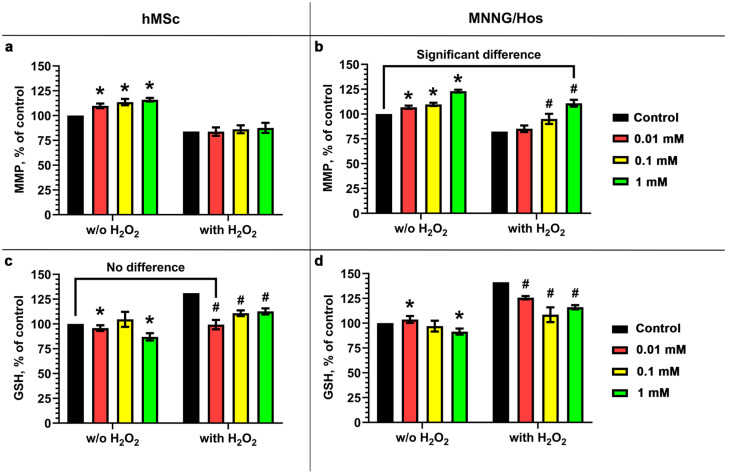
Mitochondrial membrane potential (MMP) (**a**,**b**) and GSH level (**c**,**d**) of hMSc and MNNG/Hos cells under oxidative stress induced via hydrogen peroxide in vitro. Cells were pretreated with calcein-modified CeO_2_ NPs (0.01–1 mM), then oxidative stress was induced via H_2_O_2_, and, after 24 h, the parameters were analyzed. MMP was measured using TMRE dye, and the GSH level was determined using ThiolTracker Violet. Values are given as mean percentages of test and control group scores obtained over 3 h of measurement. Data are presented as mean ± SD. Significance of differences is indicated by * for the cells not treated with H_2_O_2_ and # for the cells treated with H_2_O_2_.

**Figure 6 cells-12-02416-f006:**
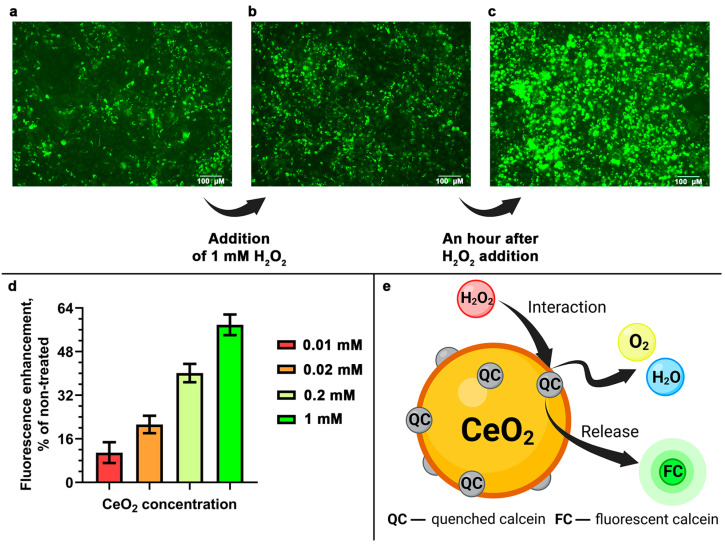
Antioxidant activity analysis of the calcein-modified CeO_2_ NPs. Intracellular calcein fluorescence in MNNG/Hos cells after 24 h of incubation with the calcein-modified CeO_2_ NPs before (**a**), immediately after (**b**), and 1 h after (**c**) H_2_O_2_ addition. H_2_O_2_-induced calcein fluorescence enhancement in MNNG/Hos cells after 24 h of preincubation with calcein-modified CeO_2_ NPs (**d**). Fluorescence enhancement data are given as a percentage of the fluorescence of the cells treated with H_2_O_2_ compared to cells not treated with H_2_O_2_. Fluorescence intensity was measured over 3 h of incubation. The proposed interaction scheme of the calcein-modified CeO_2_ NPs with H_2_O_2_ accompanied by the release of calcein with the restoration of its fluorescence (**e**).

**Figure 7 cells-12-02416-f007:**
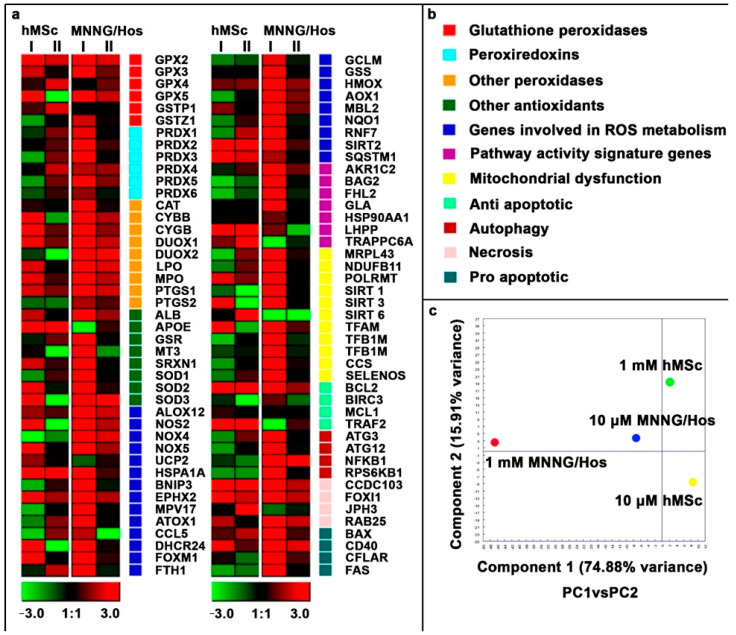
Gene expression in hMSc and MNNG/Hos cell lines after 24 h of incubation with the calcein-modified CeO_2_ NPs at various concentrations (1 mM (I) and 10 μM (II)). Data are presented as a heat map (**a**). The intensity scale for standardized gene expression values ranges from −3 (green, decreased expression) to 3 (red, increased expression) with an intermediate intensity of 1:1 (black) corresponding to control values. Cluster groups of genes with the designation of their functions and colors on the heat map (**b**). Principal component analysis (PCA) of qRT-PCR data for various cell lines and calcein-modified CeO_2_ NP concentrations (**c**).

## Data Availability

Data is contained within the article or Appendix A.

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
