# Peer review of "Calcein-Modified CeO2 for Intracellular ROS Detection: Mechanisms of Action and Cytotoxicity Analysis In Vitro"

_cells, 2023, doi:10.3390/cells12192416_

Round 1
Reviewer 1 Report
This is a review about cellular toxicity of CeO2 - calcein nanocomposite. Results of this study showed these nanoparticles’ selective cytotoxicity, inducing the death of human osteosarcoma cells and modulating the expression of key genes responsible for cell redox-21 status, proliferative and migration activity. There were some very interesting results, however, the report will benefit from revisions taking the following remarks into account.
1. The part of “Results and discussion” is less organised. It would be good to have subtitles.
2. Figure 3. Live/Dead assay. Has the experiment been repeated? How many areas/images/repeats are the subfigure c/d based on? Biological experiments need good reliability and repeatability. Please add error bars in subfigure c and d.
3. For the Gene expression (Figure 7), I cannot find how many samples or repeats for each group. Is this missing in the description or in the experimental design? Please explain.
4. Please explain/discuss more on the purpose using the two cell lines: Human mesenchymal stem cells (hMSc) and human osteosarcoma cell culture MNNG/Hos. Please also adjust the purpose of comparison of the biological endpoints between these cell lines.
General English needs to be improved. Structure of the whole report needs to be improved.
Author Response
We are very grateful for the reviewers’ comments that are aimed at improving our paper. We have thoroughly revised the manuscript in accordance with the reviewers’ comments. We have carefully checked all the points and we have tried to address all the questions and suggestions.
Reviewer #1
General comment:
This is a review about cellular toxicity of CeO2 - calcein nanocomposite. Results of this study showed these nanoparticles’ selective cytotoxicity, inducing the death of human osteosarcoma cells and modulating the expression of key genes responsible for cell redox-21 status, proliferative and migration activity. There were some very interesting results, however, the report will benefit from revisions taking the following remarks into account.
Discussion: We thank the reviewer for the positive evaluation of our work
Issue 1. The part of “Results and discussion” is less organised. It would be good to have subtitles.
Discussion: We thank the reviewer for the comment.
Changes made in the manuscript: We added subtitles in the manuscript.
Issue 2. Figure 3. Live/Dead assay. Has the experiment been repeated? How many areas/images/repeats are the subfigure c/d based on? Biological experiments need good reliability and repeatability. Please add error bars in subfigure c and d.
Discussion: We thank the reviewer for this comment. Three areas on three different microphotographs were analysed. In the revised manuscript, we added error bars (mean ± SD) in Subfigures c and d.
Changes made in the manuscript:
Figure 3. Live/Dead assay of hMSc and MNNG/Hos cells 24, 48 and 72 h after incubation with CeO2-calcein nanocomposite at various concentrations (0.01-10 mM). Cells were stained with fluorescent dyes Hoechst 33342 (stains blue the nuclei of all cells) and propidium iodide (stains red the nuclei of dead cells) (a, b), and quantitative analysis of dead cells (c, d). The IC50 line corresponds to a value of 50% dead cells. The data on с and d are presented as the mean ± SD.
Issue 3. For the Gene expression (Figure 7), I cannot find how many samples or repeats for each group. Is this missing in the description or in the experimental design? Please explain.
Discussion: We thank the reviewer for this comment. Three different samples were used, and 1 repeat per sample was performed. We added the corresponding description in the Materials and methods section (2.6. PCR-RT).
Changes made in the manuscript: Three different samples were used and 1 repeat per sample was performed.
Issue 4. Please explain/discuss more on the purpose using the two cell lines: Human mesenchymal stem cells (hMSc) and human osteosarcoma cell culture MNNG/Hos. Please also adjust the purpose of comparison of the biological endpoints between these cell lines.
Discussion: We thank the reviewer for this comment. Previously we have shown the possibility of using calcein modified ceria nanoparticles for direct monitoring of the interaction between ROS and cerium dioxide in living cells using epithelial swine testicular cell line (RSC Adv., 4(93), 51703–51710. doi:10.1039/c4ra08292c). The purpose of the present work was to clarify the molecular mechanisms of CeO2-calcein bioactivity for the further biomedical applications of this material. Therefore, we chose two types of human cells, both normal and cancer cell lines. Such a choice of cell cultures was associated with their different efficiency in nanoparticles endocytosis, as well as different intracellular metabolism and signaling pathways, which may contribute to the activity and direction of CeO2 NPs-calcein action.
Changes made in the manuscript: We added the following text at the end of the Introduction section: “The action of the nanoparticles on both normal and cancer cells was analysed in view of their possible different efficiency in endocytosis of the nanoparticles, as well as their different intracellular metabolism and signaling pathways, which may also contribute to the activity and direction of the action of the nanoparticles.”

Reviewer 2 Report
The manuscript by Popov and coworkers describes the ability of cerium oxide nanoparticles with adsorbed calcein to detect and inactivate reactive oxygen species in cells and their selective cytotoxic effect on cancer cells. The results are supported by an extensive range of biological assays and indicate that the materials may have potential therapeutic utility. Nevertheless, there are some questions that require clarification before the manuscript is suitable for publication, as summarized below.
1. The calcein-coated NPs are frequently referred to as a nanocomposite. I don’t think this is the correct use of the terminology, since these are simply NPs with a calcein surface layer, not NPs embedded in, for example, a polymer.
2. Related to point 1, section 2.1 has a heading that refers to synthesis, but no information on the preparation of the NPs is provided. The cartoon in Fig 1a clearly indicates a relatively thin layer of calcein. Details on preparation of the materials and the calcein content should be provided. Furthermore, it appears that the calcein is physically adsorbed to the particle surface. Is this the case, and what is the stability of the calcein layer under the various experimental conditions used here?
3. Section 3. Lines 200-204 should indicate that the DLS results are in Figure 1d while the UV-vis data is in Figure 1c. Additional assignment of the two UV-visible bands and the polydispersity from the DLS measurement should be provided.
4. The lengthy description of the effects of the NPs on gene expression patterns (from lines 329 to lines 461) is rather challenging to follow. The last two paragraphs of section 4 on page 14 do provide a summary of some of the main effects. Nevertheless, it would be very useful if the authors could find some additional way of highlighting the key points from the very lengthy description of the changes in gene expression.
5. Typos/grammatical errors. NPs is plural. For example, line 43 should read the CeO2 NPs exhibit a high biocompatibility…and also significantly increase their…. Please ensure that other instances of the incorrect agreement of subject/verb are corrected.
6. Please ensure that all abbreviations and acronyms are defined at first use.
noted above in author comments
Author Response
We are very grateful for the reviewers’ comments that are aimed at improving our paper. We have thoroughly revised the manuscript in accordance with the reviewers’ comments. We have carefully checked all the points and we have tried to address all the questions and suggestions.
Reviewer #2
General comment:
The manuscript by Popov and coworkers describes the ability of cerium oxide nanoparticles with adsorbed calcein to detect and inactivate reactive oxygen species in cells and their selective cytotoxic effect on cancer cells. The results are supported by an extensive range of biological assays and indicate that the materials may have potential therapeutic utility. Nevertheless, there are some questions that require clarification before the manuscript is suitable for publication, as summarized below.
Issue 1. The calcein-coated NPs are frequently referred to as a nanocomposite. I don’t think this is the correct use of the terminology, since these are simply NPs with a calcein surface layer, not NPs embedded in, for example, a polymer.
Discussion: We thank the reviewer for this comment. In the revised manuscript, we corrected the term. Where applicable, we changed the term to “calcein-modified cerium oxide nanoparticles” or similar.
Changes made in the manuscript: The terminological changes were made. We also changed the title of the manuscript.
Issue 2. Related to point 1, section 2.1 has a heading that refers to synthesis, but no information on the preparation of the NPs is provided. The cartoon in Fig 1a clearly indicates a relatively thin layer of calcein. Details on preparation of the materials and the calcein content should be provided. Furthermore, it appears that the calcein is physically adsorbed to the particle surface. Is this the case, and what is the stability layer under the various experimental conditions used here?
Discussion: We thank the reviewer for this comment. Calcein is firmly chemisorbed on the surface of CeO2 NPs due to the chelating nature of the ligand, and it can only be replaced by a ligand with a higher degree of affinity, for example, by hydrogen peroxide. Non-chelating ligands (sulfate, chloride, nitrate, etc.) cannot replace calcein, that was confirmed experimentally (Figure S1).
Changes made in the manuscript: We added the reference to the method of calcein-modified ceria nanoparticles synthesis.
The stability of the calcein layer under the various ionic microenvironments was analyzed additionally (Figure S1). It was shown that sulfate (SO42‒), chloride (Cl‒) and nitrate (NO3‒) species do not trigger calcein desorption from the surface of the CeO2 NPs. Under selected conditions, only hydrogen peroxide was able to replace calcein from the surface complex and ensure its fluorescence.
Figure S1. Analysis of the influence of various ions on the stability of the calcein-modified cerium dioxide nanoparticles.
Issue 3. Section 3. Lines 200-204 should indicate that the DLS results are in Figure 1d while the UV-vis data is in Figure 1c. Additional assignment of the two UV-visible bands and the polydispersity from the DLS measurement should be provided.
Discussion: We thank the reviewer for this comment. We have corrected the Figure 1 and changed its description in the text of the manuscript.
Changes made in the manuscript:
Figure 1. The schematic structure of calcein-modified cerium dioxide nanoparticles (a), TEM image with the corresponding selected area diffraction pattern (SAED) (b), UV-visible spectrum (c) and the hydrodynamic diameter (d) of the calcein-modified cerium dioxide nanoparticles.
Issue 4. The lengthy description of the effects of the NPs on gene expression patterns (from lines 329 to lines 461) is rather challenging to follow. The last two paragraphs of section 4 on page 14 do provide a summary of some of the main effects. Nevertheless, it would be very useful if the authors could find some additional way of highlighting the key points from the very lengthy description of the changes in gene expression.
Discussion: We thank the reviewer for this comment.
Changes made in the manuscript: In the revised manuscript, for the sake of clarity, we placed the description of each group of genes in a separate paragraph.
Issue 5. Typos/grammatical errors. NPs is plural. For example, line 43 should read the CeO2 NPs exhibit a high biocompatibility…and also significantly increase their…. Please ensure that other instances of the incorrect agreement of subject/verb are corrected.
Discussion: We thank the reviewer for this comment. We additionally checked the grammar and made the necessary corrections.
Changes made in the manuscript: We checked the manuscript for typos/errors.
Issue 6. Please ensure that all abbreviations and acronyms are defined at first use.
Discussion: We thank the reviewer for this comment. We defined all the abbreviations and acronyms at first use. Photodynamic therapy (PDT), polyvinylpyrrolidone (PVP), membrane mitochondrial potential (MMP), human mesenchymal stem cells (hMSc).
Changes made in the manuscript: Abbreviations of terms were added in the text of the manuscript when they are first mentioned.

Reviewer 3 Report
Paper entitled "CeO2 – calcein as intracellular ROS-detectable nanocomposite: Mechanisms of action and cytotoxicity analysis in vitro" authors Nikita N. Chukavin , Vladimir K. Ivanov , Anton L. Popov is a very interesting paper.
In this paper the authors presented the a new Cerium oxide (CeO2 NPs) nanoparticles as metal oxide-based nanozyme with unique reactive oxygen species (ROS) scavenging abilities. CeO2 NPs are unique inorganic antioxidants with a high degree of biocompatibility and redox activity. Additional functionalization of CeO2 NPs can give them new unique properties and possibilities, which opens up new directions for biomedical applications. Authors have studied the biocompatibility and molecular mechanisms of the CeO2-calcein nanocomposite bioactivity, which is able to simultaneously visualize and inactivate ROS in the cell, on normal and cancer cells in vitro. It has been shown that the CeO2-calcein nanocomposite has ultra-small dimensions (4-5 nm), has high colloidal stability and the ability to inactivate and simultaneously visualize the intracellular ROS location. Additionallz, authors have shown that both normal and tumor cells actively uptake the CeO2-calcein nanocomposite, meanwhile osteosarcoma cells more effectively. CeO2-calcein nanocomposite does not have cytotoxicity against normal and cancer cells in a wide range of concentrations (from 10 µM to 1 mM). However, high concentrations of the CeO2-calcein nanocomposite (10 mM) lead to cell death, and the greatest cytotoxic effect is observed in relation to cancer cells. The authors state that the cytotoxicity of the CeO2-calcein nanocomposite at high concentrations may arise due to the excessive accumulation of free calcein in cells. In particular, calcein can disrupt the ionic balance of the cell by binding Ca2+ and Mg2+ ions. It is also shown that the action of the CeO2-calcein nanocomposite simultaneously leads to an increase in MMP and a decrease in the level of GSH in cells. This effect manifests itself regardless of the experimental conditions, both w/o and with H2O2. The authors explained this by assuming that effect of the bioactivity of the nanocomposite is either not at all associated with the redox activity of CeO2 NPs, or is only partially associated with it and is primarily due to its effect on various molecular cascades in the cell.
I generally think that the published study of the bioactivity of the CeO2-calcein nanocomposite represents an innovation in the interpretation of the molecular mechanisms of the bioactivity of cerium-containing nanomaterials. Adopting defined guidelines in the interpretation of the mechanism can help develop new non-toxic theranostics to achieve a therapeutic and diagnostic effect. It has been shown that the CeO2 - calcein nanocomposite has selective cytotoxicity, inducing the death of human osteosarcoma cells and modulating the expression of key genes responsible for cell redox-status, proliferative and migration activity. Such cerium-based theranostic agent can be used in various biomedical applications.
I recommend accept paper in present form.
Author Response
We are very grateful for the reviewers’ comments that are aimed at improving our paper. We have thoroughly revised the manuscript in accordance with the reviewers’ comments. We have carefully checked all the points and we have tried to address all the questions and suggestions.
Reviewer #3
General comment:
Paper entitled "CeO2 – calcein as intracellular ROS-detectable nanocomposite: Mechanisms of action and cytotoxicity analysis in vitro" authors Nikita N. Chukavin, Vladimir K. Ivanov, Anton L. Popov is a very interesting paper.
In this paper the authors presented a new Cerium oxide (CeO2 NPs) nanoparticles as metal oxide-based nanozyme with unique reactive oxygen species (ROS) scavenging abilities. CeO2 NPs are unique inorganic antioxidants with a high degree of biocompatibility and redox activity. Additional functionalization of CeO2 NPs can give them new unique properties and possibilities, which opens up new directions for biomedical applications. Authors have studied the biocompatibility and molecular mechanisms of the CeO2-calcein nanocomposite bioactivity, which is able to simultaneously visualize and inactivate ROS in the cell, on normal and cancer cells in vitro. It has been shown that the CeO2-calcein nanocomposite has ultra-small dimensions (4-5 nm), has high colloidal stability and the ability to inactivate and simultaneously visualize the intracellular ROS location. Additionally, authors have shown that both normal and tumor cells actively uptake the CeO2-calcein nanocomposite, meanwhile osteosarcoma cells more effectively. CeO2-calcein nanocomposite does not have cytotoxicity against normal and cancer cells in a wide range of concentrations (from 10 µM to 1 mM). However, high concentrations of the CeO2-calcein nanocomposite (10 mM) lead to cell death, and the greatest cytotoxic effect is observed in relation to cancer cells. The authors state that the cytotoxicity of the CeO2-calcein nanocomposite at high concentrations may arise due to the excessive accumulation of free calcein in cells. In particular, calcein can disrupt the ionic balance of the cell by binding Ca2+ and Mg2+ ions. It is also shown that the action of the CeO2-calcein nanocomposite simultaneously leads to an increase in MMP and a decrease in the level of GSH in cells. This effect manifests itself regardless of the experimental conditions, both w/o and with H2O . The authors explained this by assuming that effect of the bioactivity of the nanocomposite is either not at all associated with the redox activity of CeO2 NPs, or is only partially associated with it and is primarily due to its effect on various molecular cascades in the cell.
I generally think that the published study of the bioactivity of the CeO2-calcein nanocomposite represents an innovation in the interpretation of the molecular mechanisms of the bioactivity of cerium-containing nanomaterials. Adopting defined guidelines in the interpretation of the mechanism can help develop new non-toxic theranostics to achieve a therapeutic and diagnostic effect. It has been shown that the CeO2 - calcein nanocomposite has selective cytotoxicity, inducing the death of human osteosarcoma cells and modulating the expression of key genes responsible for cell redox-status, proliferative and migration activity. Such cerium-based theranostic agent can be used in various biomedical applications.
I recommend accept paper in present form.
Discussion: We thank the reviewer for a positive evaluation of our manuscript.

Round 2
Reviewer 1 Report
The draft of the report is more much improved. I still have some suggestion/concerns as below.
1. Please use proper abbreviation. For example, cerium dioxide nanoparticles should be CeO2NPs.
2. Please add subtitle 3.6. If possible, separate "results“ and ”discussion".
as above.
Author Response
Reviewer #1
Issue 1: Please use proper abbreviation. For example, cerium dioxide nanoparticles should be CeO2NPs.
Discussion: We thank the reviewer for the comment.
Changes in the manuscript: We replaced «cerium dioxide nanoparticles» with «CeO2 NPs» throughout the manuscript.
Issue 2: Please add subtitle 3.6. If possible, separate "results“and ”discussion".
Discussion: We thank the reviewer for the comment. In this paper, a large number of genes was studied (about 100). Since the genes expression patterns are closely interrelated, it is more convenient to discuss the results immediately after showing the genes expression data. To address the comment of the Reviewer and to make this section more structured, we have added subtitles for each gene pattern.
Changes in the manuscript: We added subtitles in the Section 3.6.